# Cascaded Text Generation
# with Markov Transformers

**Yuntian Deng**
Harvard University
dengyuntian@seas.harvard.edu

**Alexander M. Rush**
Cornell University
arush@cornell.edu

## Abstract

The two dominant approaches to neural text generation are fully autoregressive models, using serial beam search decoding, and non-autoregressive models, using parallel decoding with no output dependencies. This work proposes an autoregressive model with sub-linear parallel time generation. Noting that conditional random fields with bounded context can be decoded in parallel, we propose an efficient cascaded decoding approach for generating high-quality output. To parameterize this cascade, we introduce a *Markov transformer*, a variant of the popular fully autoregressive model that allows us to simultaneously decode with specific autoregressive context cutoffs. This approach requires only a small modification from standard autoregressive training, while showing competitive accuracy/speed tradeoff compared to existing methods on five machine translation datasets.

## 1 Introduction

Probabilistic text generation is a ubiquitous tool in natural language processing. Originally primarily studied with respect to machine translation [1, 27], its progress has led to applications in document summarization [40, 45], data-to-text [60], image captioning [61], etc. State-of-the-art text generation approaches rely on fully autoregressive models such as RNNs and transformers [53], in which the probability of an output word depends on all previous words. At inference time, beam search is used for decoding, a left-to-right serial procedure. To speed up decoding, researchers have proposed alternative parallel generation models. One class of non-autoregressive probabilistic models assumes that each word's output probability is independent of other words [13, 67, 28]. While it is impressive that these models perform well, this independence assumption is very strong and often results in noticeable artifacts such as repetitions [13, 51].

We note that non-autoregressive models, while sufficient, are not necessary for fast probabilistic parallel generation. On parallel hardware, inference in models with bounded Markov dependencies is trivial to parallelize and requires sub-linear time w.r.t. sequence length [43, 39]. In practice, given the right parameterization, we can explore any level of autoregressive dependencies to achieve a speed/accuracy tradeoff.

In this work, we exploit this property by proposing *cascaded decoding* with a *Markov transformer* architecture. Our approach centers around a graphical model representation of the output space of text generation. Given this model, we can employ cascaded decoding [7, 8, 58, 41] for parallel text generation, using an iterative procedure that starts from a non-autoregressive model and introduces increasingly higher-order dependencies. We combine this approach with a Markov transformer, an extension to the fully autoregressive transformer architecture. This network uses barriers during training to ensure it learns fixed high-order dependencies. At test time, a single network can be used to parameterize a cascade of different graphical models. The Markov transformer only changes self-attention masks and inputs at training, and is applicable to all transformer variants.

Experiments on five machine translation datasets compare this approach to other beam search and non-autoregressive baselines. Our inference approach is comparably fast to non-autoregressive methods while allowing for local dependencies in a principled, probabilistic way. Results validate the competitive accuracy/speed tradeoff of our approach compared to existing methods. The code for reproducing all results is available at `https://github.com/harvardnlp/cascaded-generation`.

## 2 Related Work

There has been extensive interest in non-autoregressive/parallel generation approaches, aiming at producing a sequence in parallel sub-linear time w.r.t. sequence length [13, 54, 26, 67, 55, 14, 11, 12, 49, 15, 28, 16, 51, 57, 30, 42, 66, 64, 50]. Existing approaches can be broadly classified as latent variable based [13, 26, 67, 28, 42], refinement-based [25, 49, 14, 15, 11, 30, 12, 64] or a combination of both [42].

Latent-variable approaches factor out the dependencies among output words, such that we can generate each word independently of each other conditioned on those latent variables. The training of these approaches usually employs variational autoencoders, since the log marginal is intractable [21, 38, 31]. The introduced latent variables enable generation in a single forward pass, achieving $O(1)$ time complexity regardless of sequence length, but many of them suffer from generation artifacts such as repetitions [13]. While not using latent variables, our approach could be extended to incorporate them. A notable difference is that the parallel time complexity of this work is not $O(1)$ but $O(\log L)$ w.r.t. sequence length. In practice though, the only $O(\log L)$ part in our approach takes a negligible fraction of total time [51], and our approach reaches comparable speedup compared to existing approaches with $O(1)$ time complexity.

Another line of research uses refinement-based methods, where the model learns to iteratively refine a partially/fully completed hypothesis. Training usually takes the form of masked language modeling [11, 12] or imitating hand-crafted refinement policies [25, 49, 15]. Refinement-based approaches can sometimes reach better performance after multiple forward passes compared to latent variable based approaches which mostly use a single forward pass [15, 11, 42]. While our method superficially resembles refinement, our approach is probabilistic, model-based, and conceptually simpler. Training is by maximum likelihood, requires no hand-designed rules, and allows for activations to be preserved between iterations. A final benefit of our approach is that multiple lengths can be considered at no extra cost, as opposed to generating candidates under different lengths and reranking [11, 51, 28].

Our approach is motivated by structured prediction cascades (SPC) [58]. SPC is a technique in graphical models for graphical model type tasks, where we can specify the length of the sequence beforehand [58]. To the best of our knowledge, we are the first to adapt it to neural text generation. We also go beyond SPC, which uses multiple models, and show how to adapt a single Markov transformer model to learn the entire cascade. While [51] shares our motivation and combines a 0th order model with a 1st order graphical model, they do not consider higher-order models or cascades, or show how to achieve parallel sublinear time. In addition, we use a single Markov transformer to parameterize all log potentials, instead of using additional side-parameters for pairwise potentials.

## 3 Cascaded Decoding for Conditional Random Fields

Neural text decoding can be viewed as a conditional random field (CRF) [24] over a sequence of words $x_{1:L}$, where $x_i \in \mathcal{V}$ with $|\mathcal{V}| = V$, and $\mathcal{X} = \mathcal{V}^L$ is the set of all sequences. This model defines a conditional probability distribution $P(x_{1:L}|c)$, where $c$ is an arbitrary conditioning term, e.g., a source sentence. Define an $m$-th (Markov) order CRF model as,

$$P^{(m)}(x_{1:L} \mid c; \theta) \propto \exp \sum_{l=1}^{L-m} f_l^{(m)}(x_{l:l+m}, c; \theta^{(m)}),$$

where $f_l^{(m)}(\cdot)$'s are any parameterized log potentials looking at $m + 1$ words, for example local log-probabilities. For simplicity, we omit $c$ and $\theta^{(m)}$ through the rest of this paper. We can define two important special cases of this CRF model. With $m = L - 1$, we can recover fully autoregressive neural text generation models such as RNNs and transformers. Using $m = 0$ gives us non-autoregressive models.

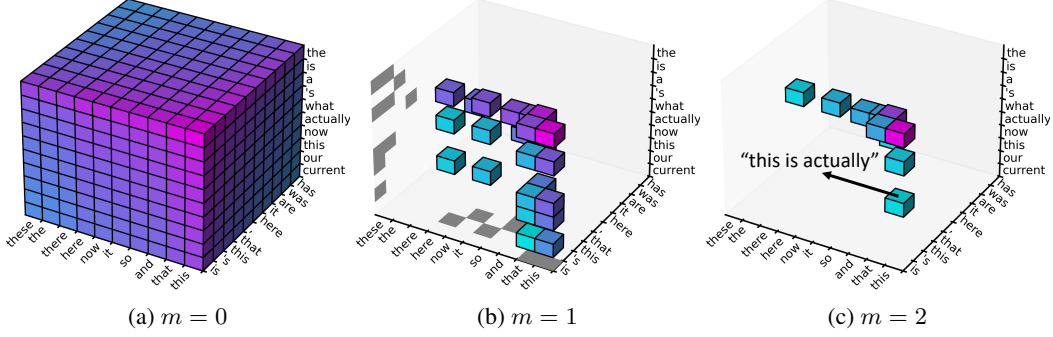

| (a) $m = 0$ | (b) $m = 1$ | (c) $m = 2$ |

Figure 1: Illustration of cascaded decoding ($K = 10$, iters $= 4$) for $\mathcal{X}_1$, $\mathcal{X}_2$, $\mathcal{X}_3$. The axes correspond to $x_1$, $x_2$ and $x_3$. (a) 0th-order (non-autoregressive) model prunes unigrams to produce $\mathcal{X}_1$; (b) 1st-order model prunes bigrams to $K$ per size-2 span (seen in 2D projection); (c) 2nd-order model prunes trigrams to $K$ total in size-3 span. Colors represent max-marginals $\text{MM}_{\mathcal{X}_m}^{(m)}(x_{1:3})$, with pink being higher and blue being lower. Fixed limit $K$ allows for efficient parallel (GPU) implementation.

Decoding aims to find the sequence with the highest model score, $\max_{x' \in \mathcal{X}} P^{(m)}(x')$. Computing this exactly can be done with the Viterbi algorithm in $O(V^{m+1}L)$; however, even for $m = 1$ this is intractable since $V$ is typically on the order of $10^4$. Beam search is commonly used instead to approximate this value, but it cannot be parallelized, and alternatives to beam search remain under-explored in the literature.

We propose an alternative *cascaded* decoding approach based on max-marginals [58], which are used as a metric to prune "unlikely" n-grams at each position based on the score of the "best" sequence with a given n-gram. To be precise, define the notation $\mathcal{X}(x_{i:j})$ to be the set of sequences that contain a span $x_{i:j}$, i.e. $\{x' \in \mathcal{X} : x'_{i:j} = x_{i:j}\}$. The max-marginal of $x_{i:j}$ is the maximum score in this set:

$$\text{MM}_{\mathcal{X}}^{(m)}(x_{i:j}) = \begin{cases} \max_{x' \in \mathcal{X}(x_{i:j})} P^{(m)}(x'_{1:L}) & \mathcal{X}(x_{i:j}) \neq \emptyset \\ 0 & \text{o.w.} \end{cases}.$$

Cascaded decoding, illustrated in Figure 1, proceeds by iteratively computing max-marginals for progressively higher-order models while filtering out unlikely spans. Starting with a complete initial set $\mathcal{X}_0 = \mathcal{X}$, for all single word spans $x_{l:l}$, we compute $M_{\mathcal{X}_0}^{(0)}$ and collect the top $K$ max-marginal values at each step to prune the search space,

$$\mathcal{X}_1 = \{x_{1:L} \in \mathcal{X}_0 : x_{l:l} \in \text{K} \arg\max_{x'_{l:l} \in \mathcal{V}^1} \text{MM}_{\mathcal{X}_0}^{(0)}(x'_{l:l}) \text{ for all } l\}.$$

We then apply a 1st order model ($m = 1$) and collect the top $K$ $x_{l:l+1}$ values with the highest max marginals $M_{\mathcal{X}_1}^{(1)}(x_{l:l+1})$ to further prune the search space,

$$\mathcal{X}_2 = \{x_{1:L} \in \mathcal{X}_1 : x_{l:l+1} \in \text{K} \arg\max_{x'_{l:l+1} \in \mathcal{V}^2} \text{MM}_{\mathcal{X}_1}^{(1)}(x'_{l:l+1}) \text{ for all } l\}.$$

We repeat the above process $M$ times with increasing $m$, and prune the search space to $\mathcal{X}_M$. It can be shown that based on properties of max marginals this set is always non-empty [58]. We decode by finding the sequence $x_{1:L}$ with the highest score $P^{(M)}(x_{1:L})$ in $\mathcal{X}_M$.

**Implementation**   The only non-parallel component of cascaded decoding is calculation of max-marginals for $m \geq 1$. With $m = 1$, max-marginals $x_{l:l+1}$ can be exactly computed using a variant of the forward-backward algorithm. This algorithm requires $O(K^2L)$ time when performed serially.

We can reduce this complexity on parallel hardware by leveraging the commutative property of $\max$ [43, 39], and computing an inside-outside prefix sum. First we pad the sequence to a power of 2 and construct a balanced binary tree with words as leaves. We then perform $\max$ operations bottom-up and top down. The height of the tree dictates the parallel time of this approach, $O(K^2 \log L)$. More details can be found in the TREEMM function in Algorithm 1, where $C_{j,k_1,k_2}^i$ is the max possible

---

**Algorithm 1** Parallel Cascaded Decoding

---

Given: max length $L$, limit $K$, log potentials $f^{(m)}$ for $m$ in $\{0, \dots, M\}$, parameters $\theta$

**function** CASCADE( )

    **for** $m = 0 \to M - 1$ **do**

        Compute potentials $f_l^{(m)}(x_{l:l+m}; \theta)$ for all $\mathcal{X}_m(x_{l:l+m}) \neq \emptyset$ $(K)$               $\triangleright O(K^2)$

        Compute first-order state relabeling $\Phi_l^{(m)}$ for all positions $l = 1 \dots L - m$     $\triangleright O(K)$

        Compute max-marginals $\text{MM}_{\mathcal{X}_m}^{(m)}$ using TREEMM                  $\triangleright O(K^2 \log L)$

        Set $\mathcal{X}_{m+1} = \left\{ x_{1:L} \in \mathcal{X}_m : x_{l:l+1} \in \underset{x'_{l:l+m} \in \mathcal{V}^{m+1}}{\text{K} \arg\max} \text{MM}_{\mathcal{X}_m}^{(m)}(x'_{l:l+m}) \text{ for all } l \right\}$    $\triangleright O(K^2)$

    **return** $\arg\max_{x' \in \mathcal{X}_M} P^{(M)}(x')$.                                           $\triangleright O(K^2 \log L)$

**function** TREEMM(First-order scores $C_{\dots}^0$ of size $L \times K \times K$)

    All $C^i, S^i, P^i$ size $2^{\log L - i} \times K \times K$, all $j \in \{1 \dots 2^{\log L - i}\}$; $P^{\log L}, S^{\log L} \leftarrow 0$

    **for** $i = 0 \to \log L - 1$ **do**              $\triangleright$ Chart max-scores computed bottom-up

        $C_{j..}^{i+1} \leftarrow \max_k C_{2j \cdot k}^i + C_{(2j+1)k \cdot}^i$

    **for** $i = \log L \to 1$ **do**           $\triangleright$ Prefix and suffix MM scores computed top-down

        $P_{2j..}^{i-1} \leftarrow P_{j..}^i$ ; $P_{2j+1..}^{i-1} \leftarrow \max_k P_{j \cdot k}^i + C_{2jk \cdot}^{i-1}$

        $S_{2j+1..}^{i-1} \leftarrow S_{j..}^i$ ; $S_{2j..}^{i-1} \leftarrow \max_k C_{(2j+1) \cdot k}^{i-1} + S_{jk \cdot}^i$

    **return** $\exp[(\max_k P_{jk \cdot}^0) + C_{j..}^0 + (\max_k S_{j \cdot k}^0)]$                        $\triangleright O(K^2 \log L)$

---

score of spans $x_{j*2^i+1:(j+1)*2^i+1}$, with the constraint of the left end being word $k_1$ and the right end being word $k_2$. We compute $C^i$ bottom-up, starting from $i = 0$ ($C^0$ is the log potentials) and merging adjacent spans in $C^i$ to get $C^{i+1}$. The prefix score $P_{j,k_1,k_2}^i$ stores the max possible score of $x_{1:j*2^i+1}$ (also with end constraints), which we compute iteratively top-down using $P^{i+1}$ and $C^i$. Similarly, the suffix score $S_{j,k_1,k_2}^i$ is the max score of $x_{(j+1)*2^i+1:}$ computed top-down. Finally, we combine the prefix scores $P^0$, suffix scores $S^0$, and log potentials $C^0$ to calculate max marginals of any edge.

For higher-order models with $m > 1$, we can compute max-marginals for $x_{l:l+m}$ using a reduction to an $m = 1$ CRF. By construction, $\mathcal{X}_m$ has exactly $K$ spans $x_{l:l+m}$ such that $\mathcal{X}(x_{l:l+m}) \neq \emptyset$ for all positions $l$. We relabel these spans $x_{l:l+m}$ as $1 \dots K$ for each position, using a mapping $\Phi_l^{(m)}(\cdots)$. This mapping implies that there are at max $K^2$ transitions between $\Phi_l^{(m)}(x_{l:l+m})$ to $\Phi_{l+1}^{(m)}(x_{l+1:l+m+1})$, resembling an $m = 1$ model over $\Phi$. Therefore, the total parallel computation cost of this process is $O(K^2 \log L)$.

The full procedure is given in Algorithm 1. As opposed to $O(V^{M+1} \log L)$ of exact search, the cascaded approximation can be computed in parallel in $O(MK^2 \log L)$. We note that this yields a sub-linear time yet (partially) autoregressive decoding algorithm.

**Handling Length** A common issue in parallel generation is the need to specify the length of the generation beforehand [13, 28]. It is hard to predict the exact length and constraining search with strict length limits the maximum achievable score. We can relax the length constraint by considering multiple lengths simultaneously. We introduce a special padding symbol pad to $\mathcal{V}$ at inference time, and add log-potentials to force pad and end-of-sentence tokens eos to transition to pad. Candidate sequences of different lengths are padded to the same length, but trailing pad's do not affect scores. The CRF parameterization allows us to consider all these lengths simultaneously, where extending the length only introduces log additional time. More details can be found at supplementary materials.

## 4 Model Parameterization: Markov Transformer

The cascaded decoding approach can be applied to any cascades of CRF models that obey the properties defined above, i.e., $m$-th order log-potentials. Given a training set $(c^j, x^j)_{1:J}$ we would

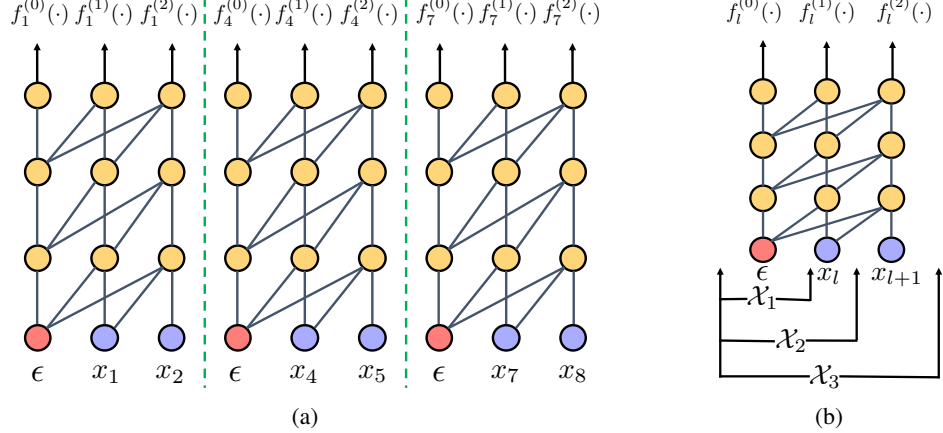

Figure 2: Markov transformer with $M = 2$ and $L = 9$. (a) At training, model state is reset with a barrier every $M + 1$ words. (b) At decoding, potential $f_l^{(0)}$ is computed at each position to get $\mathcal{X}_1$, and the dependency order is increased by introducing more columns to compute $\mathcal{X}_2$ and $\mathcal{X}_3$.

like $M + 1$ different parameters that satisfy the following MLE objectives:

$$\theta^{(m)} = \arg\max_{\theta^{(m)}} \sum_j \log P^{(m)}(x_{1:L}^j \mid c^j; \theta^{(m)}) \text{ for all } m \in \{0, \dots M\}$$

Naive approaches for cascading would require training $M + 1$ different models that are calibrated or trained together to produce similar outputs [58]. These also cannot be standard translation models such as RNNs or transformers [18, 52, 53], since they have $m = L - 1$.

We propose a training and modeling strategy to fix both of these issues. First, to reduce from $M + 1$ models to 1, we rewrite the above objective in the form:

$$(\theta^{(0)}, \dots, \theta^{(M)}) = \arg\max_{\theta^{(0)}\dots\theta^{(M)}} \frac{1}{M+1} \sum_{m=0}^{M} \sum_j \log P^{(m)}(x_{1:L}^j \mid c^j; \theta^{(m)})$$

We then make simplifying assumptions that we only want one set of model parameters $\theta$ and that the Markov order $m$ is sampled through training:

$$\theta = \arg\max_\theta \mathbb{E}_m \sum_j \log P^{(m)}(x_{1:L}^j \mid c^j; \theta)$$

In order to approximate this sampling, we train $\theta$ by starting with an autoregressive model and resetting the model's state every $M + 1$ words with a hard barrier. The first barrier is placed uniformly at random from words 1 to $M + 1$.

Next, we need a model that can be trained under this hard constraint in order to parameterize $f_l^{(m)}(\cdot)$. We propose a variant of the transformer, which we call the Markov transformer (Figure 2), that can satisfy the necessary properties. The model is trained with $(M + 1)$-spaced reset barriers with the constraint that self-attention does not cross those barriers. Transformer is particularly suited to learning with this constraint, given that it has positional encodings that encode $l$ even with explicit barriers. In order to ensure that the model can parameterize $P^{(0)}$, i.e., the prediction immediately after the barrier, we replace the first input word by a special token $\epsilon$.

To perform cascaded decoding, we simply start the computation of $f_l^{(0)}$ at each position $l$. A benefit of using a single model is that we can reuse the transformer state (neural activations) between iterations, i.e., for $f_l^{(m)}(x_{l:l+m})$ we can reuse the cached states from $f_l^{(m-1)}(x_{l:l+m-1})$. We use the output of the transformer as the log-potentials. This means each log-potential requires computing one column of the transformer, with length $m$ self-attention, requiring $O(mK)$ parallel time per iteration.

# 5 Experiments

**Datasets** We evaluate our approach on five commonly used machine translation benchmark datasets: IWSLT14 De-En [6] (~160k parallel sentences), WMT14 En-De/De-En[1] [29] (~4M parallel sentences) and WMT16 En-Ro/Ro-En[2] [3] (~610k parallel sentences). To process the data, we use Byte Pair Encoding (BPE) [46, 23] learned on the training set with a shared vocabulary between source and target. For IWSLT14 the vocabulary size is 10k; for WMT14 the vocabulary size 40k. For WMT16 we use the processed data provided by [25]. We sample all validation datasets to be at most 3k.

**Model Settings** Markov transformer uses the same hyperparameters as standard transformers. The base settings are from FAIRSEQ[3] [34]: For IWSLT14 De-En, we use 6 layers, 4 attention heads, model dimension 512, hidden dimension 1024; for WMT14 En-De/De-En and WMT16 En-Ro/Ro-En we use 6 layers, 8 attention heads, model dimension 512, hidden dimension 2048. We tie the decoder output projection matrix on all datasets [36], and we share source and target embeddings on WMT14 En-De/De-En and WMT16 En-Ro/Ro-En. It differs only in the application of attention barriers, where we set $M = 4$. The optimization settings can be found at supplementary materials.

At generation time, we predict the length $L$ using linear regression based on source length. We consider hypotheses of length $L - \Delta L$ to $L + \Delta L$ where we vary $\Delta L$ from 0 to 5. Since the Markov transformer was trained with $M = 4$, we consider applying cascaded decoding for 2 to 5 iterations (2 iterations corresponds to $M = 1$ in Algorithm 1), where more iterations consider higher local dependency orders at the cost of more computations. The limit $K$ is chosen from 16, 32, 64, 128.

**Baselines** For the fully autoregressive baseline, we use the same model setting and use beam size 5. We also compare to other parallel generation methods. These include a latent variable approach: FlowSeq [28]; refinement-based approaches: CMLM [11], Levenshtein transformer [15] and SMART [12]; a mixed approach: Imputer [42]; reinforcement learning: Imitate-NAT [57]; and another sequence-based approach: NART-DCRF [51] which combines a non-autoregressive model with a 1st-order CRF. Several of these methods use fully autoregressive reranking [13], which generally gives further improvements but requires a separate test-time model.

**Evaluation** We evaluate the BLEU score of different approaches. Following prior works [28, 51, 66], we use tokenized cased BLEU for WMT14 En-De/De-En and tokenized uncased BLEU for IWSLT14 De-En and WMT16 En-Ro/Ro-En, after removing BPE. We measure the average decoding time of a single sentence [13, 25, 16, 15, 55, 51] on a 12GB Nvidia Titan X GPU.

**Extension** Knowledge distillation [17, 19, 65] is a commonly used technique to improve the performance of parallel generation [13, 25, 28]. In knowledge distillation, we translate the training set using a fully autoregressive transformer and use the translated sentences as the new target for training.

## 5.1 Results

Results are presented in Table 1. We show the tradeoff between speedup and BLEU score by finding the configuration that gives the best BLEU score with more than $1\times$, $2\times$, ..., $7\times$ validation speedup. We presented our results in terms of the number of iterations, which is equal to $M + 1$, for comparability to refinement-based approaches.

Using knowledge distillation, our results get close to the fully autoregressive baseline: on WMT14 En-De, the gap between our approach and transformer is 0.5 BLEU, while being $2.4\times$ faster ($K = 32$, iters=5). Our results are also competitive to previous works, even those using a reranker. For example, on WMT14 En-De, we can get 26.52 BLEU score at a $4.68\times$ speedup, compared to NART-DCRF that reaches 26.80 BLEU at a $4.39\times$ speedup using 19 candidate sentences to rerank. On IWSLT14, our BLEU scores are much better than previous works: we can reach within 0.54 BLEU score compared to transformer at a $5.88\times$ speedup ($K = 16$, iters=2), 6 BLEU points better than FlowSeq.

Our approach is also competitive against previous works without distillation: at a speedup of $2.06\times$, we achieved a better BLEU score than FlowSeq-large using 30 candidates to rerank, which also has many more parameters (66M vs. 258M excluding the reranker). The one model that outperforms our approach is the Levenshtein Transformer. We note though that this model requires hand-crafted rules

Table 1: Main results. †: latency numbers not directly comparable due to platform differences.

| Approach | | Latency (Speedup) WMT14 En-De | WMT14 | | WMT16 | | IWSLT14 |
|---|---|---|---|---|---|---|---|
| Model | Settings | | En-De | De-En | En-Ro | Ro-En | De-En |
| Transformer | (beam 5) | 318.85ms ($\times 1.00$) | 27.41 | 31.49 | 33.89 | 33.82 | 34.44 |
| **With Distillation** | | | | | | | |
| Cascaded Generation | *with Speedup* | | | | | | |
| > $\times 7$ | (K=16, iters=2) | 50.28ms ($\times 6.34$) | 26.34 | 30.69 | 32.70 | 32.66 | 33.90 |
| > $\times 6/5$ | (K=32, iters=2) | 52.93ms ($\times 6.02$) | 26.43 | 30.72 | 32.73 | 32.70 | 34.01 |
| > $\times 4$ | (K=64, iters=2) | 68.09ms ($\times 4.68$) | 26.52 | 30.73 | 32.77 | 32.76 | 34.02 |
| > $\times 3$ | (K=32, iters=4) | 107.14ms ($\times 2.98$) | 26.80 | 31.22 | 33.14 | 33.22 | 34.43 |
| > $\times 2$ | (K=32, iters=5) | 132.64ms ($\times 2.40$) | 26.90 | 31.15 | 33.08 | 33.13 | 34.43 |
| > $\times 1$ | (K=64, iters=5) | 189.96ms ($\times 1.68$) | 26.92 | 31.23 | 33.23 | 33.28 | 34.49 |
| *Literature* | | | | | | | |
| FlowSeq-base [28] | | - | 21.45 | 26.16 | 29.34 | 30.44 | 27.55 |
| FlowSeq-large [28] | | - | 23.72 | 28.39 | 29.73 | 30.72 | - |
| Base CMLM[11] | (iters=10) | - | 27.03 | 30.53 | 33.08 | 33.31 | - |
| Levenshtein [15] | | 92ms ($\times 4.01$)† | 27.27 | - | - | 33.26 | - |
| SMART [12] | (iters=10) | - | 27.65 | 31.27 | - | - | - |
| Imputer [42] | (iters=1) | - | 25.8 | 28.4 | - | - | - |
| imitate-NAT [57] | | - ($\times 18.6$)† | 22.44 | 25.67 | 28.61 | 28.90 | - |
| NART-DCRF [51] | | 37ms ($\times 10.4$)† | 23.44 | 27.22 | 27.44 | - | - |
| *Literature+Reranking* | | | | | | | |
| FlowSeq-large [28] | (rescoring=30) | - | 25.31 | 30.68 | - | - | - |
| Base CMLM [11] | (iters=4, rescoring 2) | - ($\times 3.0$-$3.1$)† | 25.6-25.7 | - | - | - | - |
| imitate-NAT [57] | (rescoring=7) | - ($\times 9.70$)† | 24.15 | 27.28 | 31.45 | 31.81 | - |
| NART-DCRF [51] | (rescoring=9) | 63ms ($\times 6.14$)† | 26.07 | 29.68 | 29.99 | - | - |
| NART-DCRF [51] | (rescoring=19) | 88ms ($\times 4.39$)† | 26.80 | 30.04 | 30.36 | - | - |
| **Without Distillation** | | | | | | | |
| Cascaded Generation | *with Speedup* | | | | | | |
| > $\times 7$ | (K=16, iters=2) | 47.05ms ($\times 6.78$) | 21.34 | 26.91 | 32.11 | 32.53 | 32.95 |
| > $\times 6/5$ | (K=32, iters=2) | 54.36ms ($\times 5.87$) | 22.55 | 27.56 | 32.62 | 32.44 | 33.14 |
| > $\times 4$ | (K=64, iters=2) | 69.19ms ($\times 4.61$) | 23.09 | 27.79 | 32.78 | 32.43 | 33.25 |
| > $\times 3$ | (K=32, iters=3) | 78.29ms ($\times 4.07$) | 23.35 | 28.64 | 33.12 | 33.11 | 33.74 |
| > $\times 2/1$ | (K=64, iters=4) | 154.45ms ($\times 2.06$) | 24.40 | 29.43 | 33.64 | 33.19 | 34.08 |
| *Literature* | | | | | | | |
| FlowSeq-base [28] | | - | 18.55 | 23.36 | 29.34 | 30.44 | 24.75 |
| FlowSeq-large [28] | | - | 20.85 | 25.40 | 29.73 | 30.72 | - |
| Levenshtein [15] | | 126ms ($\times 2.93$)† | 25.20 | - | - | 33.02 | - |
| *Literature+Reranking* | | | | | | | |
| FlowSeq-large [28] | (rescoring=30) | - | 23.64 | 28.29 | 32.20 | 32.84 | - |

for training, and uses global communication, while our approach is probabilistic and only requires communicating log potentials between adjacent positions.

## 5.2 Analysis

**Candidates Searched** Unlike beam search, which is limited to a fixed number ($KL$) of candidates, cascaded search can explore an exponential number of sequences [63]. Figure 3 (a) shows the number of candidate sequences scored by cascaded decoding ($f^{(2)}$, $f^{(3)}$, $f^{(4)}$) and beam search ($f_{AR}^{(L-1)}$). We additionally note that max-marginal computations are in practice extremely fast relative to transformer computation and take less than 1% of the total time, so the bottleneck is computing potentials.

**Variable Length Generation** Cascaded decoding allows for relaxing the length constraint. Figure 3 (b) shows the effect of varying $\Delta L$ from $\{0, 3, 5\}$, where $\Delta L = 0$ corresponds to a hard length constraint, and $\Delta L = 3$ sequences of 7 possible length values from $L - 3$ to $L + 3$. By using $\Delta L = 3$, we get more than 1 BLEU improvement at any given speedup. Therefore, we use $\Delta L = 3$ for Table 1.

**Ratio of Repetitions** The independence assumption of non-autoregressive models often leads to visible artifacts in generation such as n-gram repetitions. By introducing higher-order dependencies, we can reduce the ratio of repetitions, as shown in Figure 3 (c), where we measure the extent of

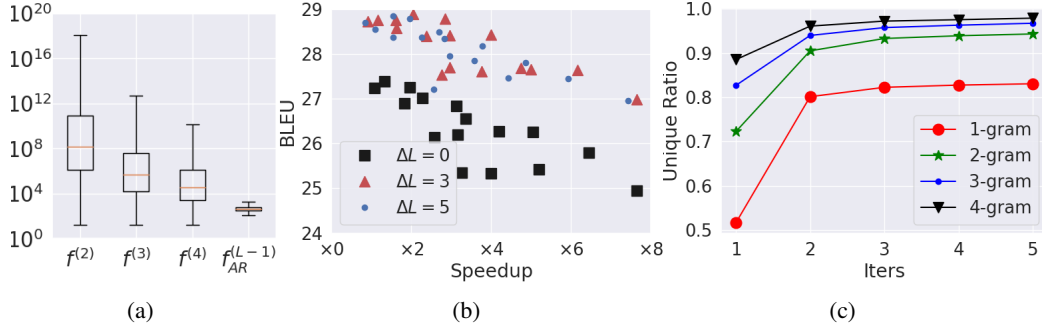

Figure 3: Analysis on WMT14 En-De val. (a) Box plot of the number of candidate sequences at different dependency orders with $K = 16$. Results include cascaded decoding with 3 iterations (scored with $f^{(2)}$), 4 iterations ($f^{(3)}$) and 5 iterations ($f^{(4)}$), and beam baseline ($f_{AR}^{(L-1)}$). (b) BLEU/speedup tradeoff as we vary $\Delta L$. The plot is drawn by varying $K$ from $\{16, 32, 64, 128\}$ and varying iterations from $\{2, 3, 4, 5\}$. (c) The ratio of n-gram repetitions evaluated using the ratio of unique n-grams as a proxy ($K = 16, \Delta L = 0$).

Table 2: Markov transformer with different search strategies on IWSLT14 De-En val w/o distillation. Column $\Delta L$ shows the length constraint ($L - \Delta L$ to $L + \Delta L$), where None denotes no constraint.

| Model | Search | Parallel | Time | $\Delta L$ | Model Score | BLEU |
|---|---|---|---|---|---|---|
| Transformer [53] | Beam (K= 5) | N | $O(KL^2)$ | None | -11.82 | 35.63 |
| Markov Trans. | Beam (K=5) | N | $O(KML)$ | None | -12.05 | 35.07 |
| | Beam (K=64) | N | - | 0 | -17.79 | 33.14 |
| | Beam (K=1024) | N | - | 0 | -16.77 | 33.33 |
| | Cascade (K=64, iters=5) | Y | - | 0 | -17.44 | 33.45 |
| | Cascade (K=64, iters=5) | Y | - | 3 | -13.87 | 35.03 |

repetitions using the ratio of unique n-grams [59]. Cascaded decoding with more than 1 iterations significantly reduces the number of repetitions.

**Markov Transformer Analysis** Table 2 shows different search algorithms for the Markov transformer. We can observe that 1) a 4th-order Markov transformer is very expressive by itself: using beam search with $K = 5$, the BLEU score (35.07) is close to the BLEU score of a transformer (35.63); 2) Cascaded decoding is less effective without distillation than serial beam search; 3) With length constraint, cascaded decoding is more effective than beam search; 4) Variable length generation can improve upon enforcing strict length constraints. Finally, we want to note that Markov transformer's complexity is lower than normal transformer, since it attends to at most $M$ past words.

**Multi-GPU** Scaling on multiple GPUs is becoming more important, given the recent trend in bigger models [47, 5]. For multi-GPU parallelization[4], each GPU takes a chunk of the sequence and forwards decoder for that chunk, while each GPU maintains full encoder states. The only communications between GPUs are the log potentials of size $L \times K \times K$ at each iteration. By using 4 GPUs, our approach can reach speedup of $2.79\times$ compared to $1.68\times$ using only 1 GPU when $K = 64$ and iters $= 5$ on WMT14 En-De test set with distillation. Note that we use batch size 1, while for most other approaches due to the global communication required between different parts of the target sentence, it is hard to reach this level of parallelism.

**Max-Marginals** To prune "unlikely" n-grams at each position, we used max-marginals instead of n-gram scores. The problem with using n-gram scores is that they do not consider compatibility with other positions. Max-marginal fixes this issue with negligible extra time. On WMT14 En-De validation set, using n-gram scores would get a BLEU score of 28.42 at 123.48ms, while using max-marginals reaches 29.24 at 128.58ms (iters $= 5, K = 32, \Delta L = 3$).

# 6    Conclusion

We demonstrate that probabilistic autoregressive models can achieve sub-linear decoding time while retaining high fidelity translations by replacing beam search with a cascaded inference approach. Our approach, based on [58], iteratively prunes the search space using increasingly higher-order models. To support this inference procedure, we utilize Markov transformers, a variant of transformer that can parameterize cascades of CRFs. Experiments on five commonly used machine translation benchmark datasets validate that our approach is competitive in terms of accuracy/speed tradeoff with other state-of-the-art parallel decoding methods, and practically useful with distillation.

Our work opens up a number of exciting future directions, such as applying this approach to longer-form text generation using latent variables, extending the Markov transformer to mimic any specified graphical model, or using more powerful globally normalized energy models instead of locally normalized ones.

## Broader Impact

Our work proposes an alternative approach to beam search that enables more efficient text generation. This work primarily uses machine translation as an application, but in the long run, it might be applied to longer-form text generation such as summarizing or translating entire documents, or be deployed to edge devices due to its faster inference and lower computational costs.

On the positive side, more efficient text generation can make these technologies more accessible to the general public. For example, machine translation can help overcome language barriers [37]; document summarization makes data more interpretable [33]. However, there are potential risks. Faster text generation has provoked concerns about generating fake news and targeted propaganda [56, 9] and might pose safety concerns if it was used to generate hate speech or to harass people [48]. Another potential problem is that it might generate language that appears fluent but fabricates facts [22].

To mitigate those issues, there have been works trying to detect machine-generated text [10, 62, 2]. While these works address some concerns over the abuse of text generation, we should be cautious that fake news detection is still a mostly unsolved technical problem and requires active future research [44, 4] as well as non-technical mitigation efforts.

## Acknowledgments and Disclosure of Funding

We would like to thank Justin Chiu, Demi Guo, Yoon Kim, David Rosenberg, Zachary Ziegler, and Jiawei Zhou for helpful feedback. This project was supported by NSF SHF 1704834, CAREER IIS-1845664, and Intel. YD is supported by a Baidu AI fellowship.

## Footnotes

[1] http://www.statmt.org/wmt14/translation-task.html

[2] http://www.statmt.org/wmt16/translation-task.html

[3] https://github.com/pytorch/fairseq/tree/master/examples/translation

[4]We use `https://pytorch.org/docs/stable/multiprocessing.html`.

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
