[Supplementary Material]

# Supplementary Materials for
# Cascaded Text Generation with Markov Transformers

## Appendix A: Cascaded Decoding Examples

We show a decoding example in Table 3 ($K = 5$, $\Delta L = 1$, iters=5). We sort states by max-marginals in descending order and use - to denote invalid states (with $-\infty$ log max-marginals). In this simple sentence, using 1 iteration ($m = 0$, non-autoregressive model) repeats the word "woman" ($m = 0$, first row, $x_{4:4+m}$). Introducing higher order dependencies fixes this issue.

Table 3: Cascaded Decoding Example. When $m = 4$, Viterbi in $\mathcal{X}_4$ returns "an amazing woman . eos". The source is "eine erstaunliche frau . eos" and the target is "an amazing woman . eos".

| $m$ | $x_{1:1+m}$ | $x_{2:2+m}$ | $x_{3:3+m}$ | $x_{4:4+m}$ | $x_{5:5+m}$ | $x_{6:6+m}$ | $x_{7:7+m}$ | $x_8$ |
|---|---|---|---|---|---|---|---|---|
| | an | amazing | woman | woman | eos | eos | eos | pad |
| | amazing | woman | amazing | . | . | pad | pad | - |
| 0 | incredible | an | an | amazing | woman | . | . | - |
| | this | remarkable | . | eos | amazing | woman | woman | - |
| | remarkable | incredible | women | an | women | women | women | - |
| | an amazing | amazing woman | woman . | . eos | eos pad | pad pad | pad pad | |
| | an incredible | incredible woman | amazing woman | woman . | . eos | eos pad | eos pad | |
| 1 | this amazing | remarkable woman | women . | amazing woman | woman . | . eos | - | |
| | an remarkable | woman amazing | woman woman | . . | women . | woman eos | - | |
| | amazing woman | amazing women | an amazing | . woman | . . | - | - | |
| | an amazing woman | amazing woman . | woman .eos | . eos pad | eos pad pad | pad pad pad | | |
| | an incredible woman | incredible woman . | women . eos | woman . eos | . eos pad | eos pad pad | | |
| 2 | this amazing woman | remarkable woman . | woman woman . | . . eos | woman . eos | . eos pad | | |
| | an remarkable woman | amazing women . | woman . . | . woman . | . . eos | - | | |
| | an amazing women | amazing woman woman | woman . woman | woman . . | - | - | | |
| | an amazing woman . | amazing woman . eos | woman . eos pad | . eos pad pad | eos pad pad pad | | | |
| | an incredible woman . | incredible woman . eos | women . eos pad | woman . eos pad | . eos pad pad | | | |
| 3 | this amazing woman . | remarkable woman . eos | woman woman . eos | . . eos pad | woman . eos pad | | | |
| | an remarkable woman . | amazing women . eos | woman . . eos | . woman . eos | . . eos pad | | | |
| | an amazing women . | amazing woman woman . | woman . woman . | woman . . eos | - | | | |

Table 4: Cascaded Decoding Example. When $m = 4$, Viterbi in $\mathcal{X}_4$ returns "what has happened ? eos". The source is "was ist passiert ? eos" and the target is "what happened ? eos".

| $m$ | $x_{1:1+m}$ | $x_{2:2+m}$ | $x_{3:3+m}$ | $x_{4:4+m}$ | $x_{5:5+m}$ | $x_{6:6+m}$ | $x_{7:7+m}$ | $x_8$ |
|---|---|---|---|---|---|---|---|---|
| | what | happened | happened | ? | eos | eos | eos | pad |
| | so | has | ? | eos | ? | pad | pad | - |
| 0 | now | did | what | happened | happened | ? | ? | - |
| | and | what | happen | happen | happen | happened | . | - |
| | well | 's | eos | happens | happens | . | happened | - |
| | what has | has happened | happened ? | ? eos | eos pad | pad pad | pad pad | |
| | so what | what happened | what happened | happened ? | ? eos | eos pad | eos pad | |
| 1 | and what | 's happened | happen ? | happens ? | happens ? | ? eos | - | |
| | what 's | did what | what ? | happen ? | happen ? | . eos | - | |
| | now what | did happened | what happens | ? ? | happened ? | happened eos | - | |
| | what has happened | has happened ? | happened ? eos | ? eos pad | eos pad pad | pad pad pad | | |
| | so what happened | what happened ? | happened ? ? | ? ? eos | ? eos pad | eos pad pad | | |
| 2 | what 's happened | 's happened ? | happen ? ? | happen ? eos | happened ? eos | happened eos pad | | |
| | and what happened | did what ? | happen ? eos | happens ? eos | happen ? eos | . eos pad | | |
| | now what happened | did what happened | what happened ? | happened ? eos | happens ? eos | ? eos pad | | |
| | what has happened ? | has happened ? eos | happened ? eos pad | ? eos pad pad | eos pad pad pad | | | |
| | so what happened ? | what happened ? eos | happened ? ? | ? ? eos pad | ? eos pad pad | | | |
| 3 | and what happened ? | 's happened ? eos | what happened ? eos | happened ? eos pad | happens ? eos pad | | | |
| | what 's happened ? | has happened ? ? | happen ? eos pad | happens ? eos pad | happen ? eos pad | | | |
| | now what happened ? | what happened ? ? | happen ? ? eos | happen ? eos pad | happened ? eos pad | | | |

In Tables 4, 5, 6, 7, 8 we show more examples from IWSLT14 De-En val.

Table 5: Cascaded Decoding Example. When $m = 4$, Viterbi in $\mathcal{X}_4$ returns "you 're happy . eos". The source is "du bist glücklich . eos" and the target is "you 're happy . eos".

| $m$ | $x_{1:1+m}$ | $x_{2:2+m}$ | $x_{3:3+m}$ | $x_{4:4+m}$ | $x_{5:5+m}$ | $x_{6:6+m}$ | $x_{7:7+m}$ | $x_8$ |
|---|---|---|---|---|---|---|---|---|
|  | you | 're | happy | . | eos | eos | eos | pad |
|  | happy | are | lucky | eos | . | pad | pad | - |
| 0 | your | you | gla@@ | happy | happy | . | . | - |
|  | and | 's | good | lucky | ? | happy | happy | - |
|  | i | be | fortun@@ | ful | you | ? | ? | - |
|  | you 're | 're happy | happy . | . eos | eos pad | pad pad | pad pad |  |
|  | you are | are happy | lucky . | . . | . eos | eos pad | eos pad |  |
| 1 | you be | are lucky | good . | happy . | happy . | . eos | - |  |
|  | you 's | be happy | happy happy | ful . | ? eos | ? eos | - |  |
|  | and you | 're lucky | happy ful | lucky . | you . | happy eos | - |  |
|  | you 're happy | 're happy . | happy . eos | . eos pad | eos pad pad | pad pad pad |  |  |
|  | you are happy | are happy . | lucky . eos | . . eos | . eos pad | eos pad pad |  |  |
| 2 | you be happy | be happy . | happy . . | happy . eos | you . eos | happy eos pad |  |  |
|  | you 're lucky | 're lucky . | happy happy . | ful . eos | ? eos pad | ? eos pad |  |  |
|  | you are lucky | are lucky . | happy ful . | lucky . eos | happy . eos | . eos pad |  |  |
|  | you 're happy . | 're happy . eos | happy . eos pad | . eos pad pad | eos pad pad pad |  |  |  |
|  | you are happy . | are happy . eos | lucky . eos pad | . . eos pad | . eos pad pad |  |  |  |
| 3 | you be happy . | be happy . eos | happy . . eos | lucky . eos pad | happy . eos pad |  |  |  |
|  | you 're lucky . | 're lucky . eos | happy ful . eos | ful . eos pad | ? eos pad pad |  |  |  |
|  | you are lucky . | are lucky . eos | happy happy . eos | happy . eos pad | you . eos pad |  |  |  |

Table 6: Cascaded Decoding Example. When $m = 4$, Viterbi in $\mathcal{X}_4$ returns "let 's move . eos". The source is "bewe@@ g dich . eos" and the target is "move it . eos".

| $m$ | $x_{1:1+m}$ | $x_{2:2+m}$ | $x_{3:3+m}$ | $x_{4:4+m}$ | $x_{5:5+m}$ | $x_{6:6+m}$ | $x_{7:7+m}$ | $x_8$ |
|---|---|---|---|---|---|---|---|---|
|  | move | move | . | eos | eos | eos | eos | pad |
|  | let | . | eos | . | . | pad | pad | - |
| 0 | so | moving | move | ? | ? | . | . | - |
|  | just | 's | forward | forward | here | ? | ? | - |
|  | now | let | moving | it | forward | here | here | - |
|  | let 's | 's move | move . | . eos | eos pad | pad pad | pad pad |  |
|  | just move | 's moving | moving . | it . | . eos | eos pad | eos pad |  |
| 1 | so move | move forward | move it | forward . | here . | . eos | - |  |
|  | move . | . forward | move forward | ? eos | ? eos | ? eos | - |  |
|  | move 's | . moving | move ? | . . | forward . | - | - |  |
|  | let 's move | 's move . | move . eos | . eos pad | eos pad pad | pad pad pad |  |  |
|  | let 's moving | 's move it | move it . | it . eos | . eos pad | eos pad pad |  |  |
| 2 | move 's move | 's move forward | move forward . | forward . eos | ? eos pad | ? eos pad |  |  |
|  | move . moving | 's moving . | moving . eos | ? eos pad | here . eos | . eos pad |  |  |
|  | move 's moving | 's move ? | move ? eos | . . eos | - | - |  |  |
|  | let 's move . | 's move . eos | move . eos pad | . eos pad pad | eos pad pad pad |  |  |  |
|  | let 's move it | 's move it . | move it . eos | it . eos pad | . eos pad pad |  |  |  |
| 3 | let 's moving . | 's moving . eos | moving . eos pad | forward . eos pad | here . eos pad |  |  |  |
|  | let 's move forward | 's move forward . | move forward . eos | ? eos pad pad | ? eos pad pad |  |  |  |
|  | let 's move ? | 's move ? eos | move ? eos pad | . . eos pad | - |  |  |  |

Table 7: Cascaded Decoding Example. When $m = 4$, Viterbi in $\mathcal{X}_4$ returns "very , very hard . eos". The source is "sehr sehr schwer . eos" and the target is "very very hard . eos".

| $m$ | $x_{1:1+m}$ | $x_{2:2+m}$ | $x_{3:3+m}$ | $x_{4:4+m}$ | $x_{5:5+m}$ | $x_{6:6+m}$ | $x_{7:7+m}$ | $x_8$ |
|---|---|---|---|---|---|---|---|---|
| 0 | very | difficult | difficult | . | eos | eos | eos | pad |
| | it | hard | hard | eos | . | pad | pad | - |
| | really | very | . | difficult | difficult | . | . | - |
| | extremely | tough | very | hard | hard | difficult | difficult | - |
| | that | , | tough | very | very | hard | hard | - |
| 1 | very , | , very | very difficult | difficult . | . eos | eos pad | pad pad | |
| | very very | very hard | very hard | hard . | eos pad | pad pad | eos pad | |
| | really , | very difficult | hard . | . eos | difficult . | . eos | - | |
| | it very | , hard | difficult . | hard eos | hard . | difficult eos | - | |
| | extremely , | , difficult | tough . | difficult eos | . . | hard eos | - | |
| 2 | very , very | , very hard | very hard . | hard . eos | . eos pad | eos pad pad | | |
| | very very difficult | , very difficult | very difficult . | difficult . eos | eos pad pad | pad pad pad | | |
| | very very hard | very difficult . | difficult . eos | . eos pad | . . eos | . eos pad | | |
| | really , very | very hard . | hard . eos | hard eos pad | hard . eos | hard eos pad | | |
| | it very difficult | , hard . | very hard eos | difficult eos pad | difficult . eos | difficult eos pad | | |
| 3 | very , very hard | , very hard . | very hard . eos | hard . eos pad | . eos pad pad | | | |
| | very , very difficult | , very difficult . | very difficult . eos | difficult . eos pad | eos pad pad pad | | | |
| | very very difficult . | very difficult . eos | difficult . eos pad | . eos pad pad | difficult . eos pad | | | |
| | very very hard . | very hard . eos | hard . eos pad | hard eos pad pad | hard . eos pad | | | |
| | really , very hard | , very hard eos | very hard eos pad | difficult eos pad pad | . . eos pad | | | |

Table 8: Cascaded Decoding Example. When $m = 4$, Viterbi in $\mathcal{X}_4$ returns "the opposite thing happened . eos". The source is "das gegenteil passierte . eos" and the target is "the opposite happened . eos".

| $m$ | $x_{1:1+m}$ | $x_{2:2+m}$ | $x_{3:3+m}$ | $x_{4:4+m}$ | $x_{5:5+m}$ | $x_{6:6+m}$ | $x_{7:7+m}$ | $x_8$ |
|---|---|---|---|---|---|---|---|---|
| 0 | the | opposite | opposite | happened | eos | eos | eos | pad |
| | and | contr@@ | thing | was | . | pad | pad | - |
| | so | other | ary | thing | happened | . | . | - |
| | but | the | happened | did | happening | happened | happened | - |
| | well | conver@@ | was | opposite | happen | happen | happen | - |
| 1 | the opposite | opposite thing | thing happened | happened . | . eos | eos pad | pad pad | |
| | the contr@@ | contr@@ ary | ary happened | was happening | happening . | . eos | eos pad | |
| | and the | the opposite | opposite happened | thing happened | happened . | pad pad | - | |
| | the other | other thing | thing was | did . | eos pad | . happened eos | - | |
| | so the | opposite opposite | was happened | was happened | . . | - | - | |
| 2 | the opposite thing | opposite thing happened | thing happened . | happened . eos | . eos pad | eos pad pad | | |
| | the contr@@ ary | contr@@ ary happened | ary happened . | was happening . | happening . eos | . eos pad | | |
| | and the opposite | the opposite happened | opposite happened . | was happened . | happened . eos | happened eos pad | | |
| | the other thing | other thing happened | thing was happening | happened . . | . . eos | pad pad pad | | |
| | so the opposite | opposite thing was | thing was happened | thing happened . | - | - | | |
| 3 | the opposite thing happened | opposite thing happened . | thing happened . eos | happened . eos pad | . eos pad pad | | | |
| | the contr@@ ary happened | contr@@ ary happened . | ary happened . eos | was happening . eos | happening . eos pad | | | |
| | and the opposite happened | the opposite happened . | opposite happened . eos | was happened . eos | happened . eos pad | | | |
| | the other thing happened | other thing happened . | thing was happening . | happened . . eos | . . eos pad | | | |
| | the opposite thing was | opposite thing was happening | thing was happened . | - | - | | | |

## Appendix B: More Visualizations

Figure 4: Illustration of cascaded decoding ($K = 10$, iters=4) for $\mathcal{X}_1, \mathcal{X}_2, \mathcal{X}_3$.

Figure 5: Illustration of cascaded decoding ($K = 10$, iters=4) for $\mathcal{X}_1, \mathcal{X}_2, \mathcal{X}_3$.

We include more visualizations of $\mathcal{X}_1$, $\mathcal{X}_2$ and $\mathcal{X}_3$ in Figure 4 and Figure 5. These examples are taken from IWSLT14 De-En val.

## Appendix C: Variable Length Generation Potentials

To handle length, we introduce an additional padding symbol `pad` to $\mathcal{V}$, and change the log potentials to enforce the considered candidates are of length $L - \Delta L$ to $L + \Delta L$. Note that we can only enforce that for $m \geq 1$, and for $m = 0$ we manually add `pad` to the pruned vocabulary.

We start cascaded search using a sequence of length $L + \Delta L + 1$. The main ideas are: 1) We make `eos` and `pad` to always transition to `pad` such that sequences of different lengths can be compared; 2) We disallow `eos` to appear too early or too late to satisfy the length constraint; 3) We force the last token to be `pad` such that we don't end up with sentences without `eos` endings. Putting these ideas together, the modified log potentials we use are:

$$f'^{(m)}_l(x_{l:l+m})$$

$$= \begin{cases} 0, \text{ if } x_{l+m-1} = \texttt{eos} \wedge x_{l+m} = \texttt{pad} \\ -\infty, \text{ if } x_{l+m-1} = \texttt{eos} \wedge x_{l+m} \neq \texttt{pad} \, (\texttt{eos} \to \texttt{pad}) \\ 0, \text{ if } x_{l+m-1} = \texttt{pad} \wedge x_{l+m} = \texttt{pad} \\ -\infty, \text{ if } x_{l+m-1} = \texttt{pad} \wedge x_{l+m} \neq \texttt{pad} \, (\texttt{pad} \to \texttt{pad}) \\ -\infty, \text{ if } x_{l+m-1} \neq \texttt{pad} \wedge x_{l+m-1} \neq \texttt{eos} \wedge x_{l+m} = \texttt{pad} \, (\text{nothing else} \to \texttt{pad}) \\ -\infty, \text{ if } l+m < L - \Delta L \wedge x_{l+m} = \texttt{eos} \, (\texttt{eos} \text{ cannot appear too early}) \\ 0, \text{ if } l+m = L + \Delta L + 1 \text{ and } x_{l+m} = \texttt{pad} \\ -\infty, \text{ if } l+m = L + \Delta L + 1 \text{ and } x_{l+m} \neq \texttt{pad} \, (\text{the last token must be } \texttt{pad}) \\ f^{(m)}_l(x_{l:l+m}), \text{ o.t.} \end{cases} \quad .$$

Note that we only considered a single sentence above, but batching is straightforward to implement and we refer interested readers to our code[5] for batch implementations.

## Appendix D: Full Results

In the main experiment table we showed latency/speedup results for WMT14 En-De. In Table 9, Table 10, Table 11 and Table 12 we show the latency/speedup results for other datasets. Same as in the main experiment table, we use the validation set to choose the configuration with the best BLEU score under speedup $> \times 1$, $> \times 2$, etc.

Table 9: Results on WMT14 De-En.

| Model | Settings | Latency (Speedup) | BLEU |
|---|---|---|---|
| Transformer | (beam 5) | 294.64ms ($\times 1.00$) | 31.49 |
| **With Distillation** | | | |
| Cascaded Generation *with Speedup* | | | |
| $> \times 7$ | (K=16, iters=2) | 43.41ms ($\times 6.79$) | 30.69 |
| $> \times 6$ | (K=32, iters=2) | 52.06ms ($\times 5.66$) | 30.72 |
| $> \times 5$ | (K=16, iters=3) | 62.06ms ($\times 4.75$) | 30.96 |
| $> \times 4/3$ | (K=32, iters=3) | 79.01ms ($\times 3.73$) | 31.08 |
| $> \times 2/1$ | (K=32, iters=5) | 129.67ms ($\times 2.27$) | 31.15 |
| **Without Distillation** | | | |
| Cascaded Generation *with Speedup* | | | |
| $> \times 6/5$ | (K=32, iters=2) | 53.83ms ($\times 5.47$) | 27.56 |
| $> \times 4$ | (K=32, iters=3) | 81.10ms ($\times 3.63$) | 28.64 |
| $> \times 3$ | (K=32, iters=4) | 106.97ms ($\times 2.75$) | 28.73 |
| $> \times 2$ | (K=64, iters=4) | 154.15ms ($\times 1.91$) | 29.43 |
| $> \times 1$ | (K=128, iters=4) | 269.59ms ($\times 1.09$) | 29.66 |

Table 10: Results on WMT16 En-Ro.

| Model | Settings | Latency (Speedup) | BLEU |
|---|---|---|---|
| Transformer | (beam 5) | 343.28ms ($\times$1.00) | 33.89 |
| **With Distillation** | | | |
| Cascaded Generation *with Speedup* | | | |
| > $\times$7 | (K=16, iters=2) | 49.38ms ($\times$6.95) | 32.70 |
| > $\times$6 | (K=32, iters=2) | 54.56ms ($\times$6.29) | 32.73 |
| > $\times$5 | (K=16, iters=3) | 66.33ms ($\times$5.18) | 32.89 |
| > $\times$4 | (K=32, iters=3) | 77.39ms ($\times$4.44) | 33.16 |
| > $\times$3 | (K=64, iters=3) | 108.57ms ($\times$3.16) | 33.23 |
| > $\times$2 | (K=64, iters=4) | 142.23ms ($\times$2.41) | 33.30 |
| > $\times$1 | (K=64, iters=5) | 179.07ms ($\times$1.92) | 33.23 |
| **Without Distillation** | | | |
| Cascaded Generation *with Speedup* | | | |
| > $\times$7 | (K=16, iters=2) | 45.18ms ($\times$7.60) | 32.11 |
| > $\times$6 | (K=32, iters=2) | 51.38ms ($\times$6.68) | 32.62 |
| > $\times$5 | (K=16, iters=3) | 60.34ms ($\times$5.69) | 32.67 |
| > $\times$4 | (K=32, iters=3) | 73.99ms ($\times$4.64) | 33.12 |
| > $\times$3 | (K=64, iters=3) | 105.46ms ($\times$3.26) | 33.48 |
| > $\times$2 | (K=64, iters=4) | 145.18ms ($\times$2.36) | 33.64 |
| > $\times$1 | (K=128, iters=5) | 325.42ms ($\times$1.05) | 33.52 |

Table 11: Results on WMT16 Ro-En.

| Model | Settings | Latency (Speedup) | BLEU |
|---|---|---|---|
| Transformer | (beam 5) | 318.57ms ($\times$1.00) | 33.82 |
| **With Distillation** | | | |
| Cascaded Generation *with Speedup* | | | |
| > $\times$6/5 | (K=16, iters=2) | 46.84ms ($\times$6.80) | 32.66 |
| > $\times$4 | (K=16, iters=3) | 62.57ms ($\times$5.09) | 33.00 |
| > $\times$3 | (K=16, iters=5) | 99.25ms ($\times$3.21) | 33.04 |
| > $\times$2 | (K=64, iters=3) | 103.85ms ($\times$3.07) | 33.17 |
| > $\times$1 | (K=64, iters=5) | 181.18ms ($\times$1.76) | 33.28 |
| **Without Distillation** | | | |
| Cascaded Generation *with Speedup* | | | |
| > $\times$6 | (K=16, iters=2) | 47.58ms ($\times$6.70) | 32.53 |
| > $\times$5 | (K=32, iters=2) | 54.05ms ($\times$5.89) | 32.44 |
| > $\times$4 | (K=16, iters=3) | 60.94ms ($\times$5.23) | 33.00 |
| > $\times$3 | (K=32, iters=4) | 100.29ms ($\times$3.18) | 33.10 |
| > $\times$2 | (K=64, iters=3) | 105.21ms ($\times$3.03) | 33.22 |
| > $\times$1 | (K=128, iters=4) | 282.76ms ($\times$1.13) | 33.29 |

Table 12: Results on IWSLT14 De-En.

| Model | Settings | Latency (Speedup) | BLEU |
|---|---|---|---|
| Transformer | (beam 5) | 229.76ms ($\times$1.00) | 34.44 |
| **With Distillation** | | | |
| Cascaded Generation *with Speedup* | | | |
| $>$ $\times$6/5 | (K=16, iters=2) | 39.38ms ($\times$5.83) | 33.90 |
| $>$ $\times$4 | (K=32, iters=3) | 60.27ms ($\times$3.81) | 34.33 |
| $>$ $\times$3 | (K=32, iters=4) | 78.27ms ($\times$2.94) | 34.43 |
| $>$ $\times$2/1 | (K=64, iters=5) | 117.90ms ($\times$1.95) | 34.49 |
| **Without Distillation** | | | |
| Cascaded Generation *with Speedup* | | | |
| $>$ $\times$5 | (K=64, iters=2) | 48.59ms ($\times$4.73) | 33.25 |
| $>$ $\times$4 | (K=32, iters=3) | 60.09ms ($\times$3.82) | 33.74 |
| $>$ $\times$3 | (K=64, iters=3) | 75.64ms ($\times$3.04) | 33.96 |
| $>$ $\times$2 | (K=64, iters=5) | 121.95ms ($\times$1.88) | 34.08 |
| $>$ $\times$1 | (K=128, iters=5) | 189.10ms ($\times$1.22) | 34.15 |

## Appendix E: Optimization Settings

Table 13: Optimization settings. We use the same settings for knowledge distillation experiments.

| Dataset | dropout | fp16 | GPUs | batch | accum | warmup steps | max steps | max lr | weight decay |
|---|---|---|---|---|---|---|---|---|---|
| WMT14 En-De/De-En | 0.1 | Y | 3 | 4096 | 3 | 4k | 240k | 7e-4 | 0 |
| WMT16 En-Ro/Ro-En | 0.3 | Y | 3 | 5461 | 1 | 10k | 240k | 7e-4 | 1e-2 |
| IWSLT14 De-En | 0.3 | N | 1 | 4096 | 1 | 4k | 120k | 5e-4 | 1e-4 |

Our approach is implemented in PyTorch [35], and we use 16GB Nvidia V100 GPUs for training. We used Adam optimizer [20], with betas 0.9 and 0.98. We use inverse square root learning rate decay after warmup steps [34]. We train with label smoothing strength 0.1 [32]. For model selection, we used BLEU score on validation set. For Markov transformers, we use cascaded decoding with $K = 16$ and $\Delta L = 3$ to compute validation BLEU score. Other hyperparameters can be found at Table 13.

## Footnotes

[5]`https://github.com/harvardnlp/cascaded-generation`