[Reviews · NeurIPS 2020]

Review 1

Summary and Contributions: This paper presents an approach to non-autoregressive sequence generation using a cascade of transformers. They formulate transformers as CRFs, where an mth-order transformer depends on the previous m words in the decoder (as well as all words in the input). Critically, by pruning the state space at each each timestep using lower-order models, inference in higher-order models becomes faster. This pruning is based on max marginals: the score of the best path using a particular word (for 0th order models) or n-gram (for higher-order). These max marginals themselves would pose a O(L) serial bottleneck, except that the paper presents a clever parallel algorithm for computing them in O(log L) time. Results on 5 MT settings show that performance is nearly as good as an autoregressive transformer, better than most past non-autoregressive models, and giving a 3x-5x speedup without much loss in performance compared to the autoregressive model. Distillation from the autoregressive model is necessary to achieve this strong performance. The authors have released their model and hyperparameters for all experiments.

Strengths: This paper has some very nice algorithmic contributions. I am not sure whether this will be a preferred method for non-autoregressive generation (see below in the Weaknesses), but I like what the authors are trying to do. Coarse-to-fine pruning using max marginals is a very appealing technique that hasn't really seen the light of day in the neural era. It's impressive to me that the authors have gotten this as fast as it is, and there are some strong algorithmic components to make this work. In particular, the parallel max marginal computation is pretty cool. The paper introduces a significantly different way of doing things than most other non-autoregressive techniques, so I think it will be a valuable reference and inspiration to other researchers who want to carry on in this direction. The model is also nicely tuned. Folding all of the models into a single one is reasonably elegant. Between this and the knowledge distillation, the authors seem to have put a fair amount of effort into making this work well, and this effort pays off nicely in the results. The results themselves seem quite strong, handily outperforming many of the past non-autoregressive models with a decent speedup.

Weaknesses: While I am advocating for this paper's acceptance, I'm curious as to whether the authors think this will truly be the dominant approach going forward in this area. I find this approach theoretically more appealing than the Levenshtein transformer, but I think the "global communication" as a negative feature of that model isn't strictly a negative. Sure, the more local nature of this one gives a speedup. But successfully capturing long-range dependencies is one of the things transformer models like GPT-3 seem to be good at. This is a limitation of the paper only evaluating on MT; in MT, the input heavily constrains the shape of the output and long-range output dependencies may not be quite as necessary. But in tasks like summarization or open-ended text generation like story generation, the input governs the shape out of the output much less and I wonder whether this more local model is sufficient. It seems like while approaches like the Levenshtein transformer and other insertion-based approaches maybe aren't as well motivated and are harder to get working well, a good generation order there could outperform the approach in this work for more complex generation settings. Another mild weakness is the need for the length prediction with linear regression. Again, this works for MT, but I wonder about other settings.

Correctness: This paper is very nice empirically and supports its claims well.

Clarity: Yes, this paper is very clear.

Relation to Prior Work: The set of work cited is impressive and the comparisons in the experiments are thorough as far as I can tell.

Reproducibility: Yes

Additional Feedback: Line 100: "wih" =============== Post-response: thanks for the nice response. I do take your point about length prediction being an upper bound. Otherwise, my review and score is unchanged -- I still think this is solid work.


Review 2

Summary and Contributions: The paper proposes a model for sub-linear time, semi-autoregressive neural text generation with a model that considers Markov dependencies in the output. The model is based on adapting the framework on structured prediction cascades, where candidate outputs are pruned during decoding through scoring with CRF models of increasing context length (0- to Mth-order dependencies in the outputs). The max marginals of each n-gram are computed to filter the top-K candidates at each step with a parallelized implementation. The model is a Transformer with hard barriers between every M+1 words during training to enforce the Markov dependency structure. The results show that up to 7 time speed-ups over fully autoregressive decoding can be obtained at little loss in translation accuracy when knowledge distillation is used, which compared favorably to most previous non-autoregressive approaches.

Strengths: - The paper proposes a principled approach to semi-autoregressive generation with a novel application of previous work on structured prediction in a framework with more expressive neural models. - The approach provides a good trade-off between speed and accuracy compared to previous approaches. Extensive model analysis is included.

Weaknesses: - The description of the approach is complete and clear, but quite dense, in particular for cascaded decoding. So I think the paper could be improved by having more high-level explanations or intuitions. - The paper presents a general and powerful framework, but the model is then trained so that the log potential of an n-gram is the next word probability, which makes sense for training efficiency but seems to not fully exploit the expressive potential.

Correctness: The claims and empirical methodology are correct.

Clarity: The paper is well written.

Relation to Prior Work: All relevant prior work is mentioned, but there should preferably be more discussion on the relation to the closest previous work on semi-autoregressive generation, in particular Sun et al., 2019, who also proposed a CRF-based approach.

Reproducibility: Yes

Additional Feedback: What is the effect of computing the actual max marginals in decoding vs approximating it by only using the n-gram score? In the result tables, it may improve clarity and consistency to use M instead of "iterations". --- Thanks for the response and clarifications.


Review 3

Summary and Contributions: This paper proposes a new non-autoregressive text generation algorithm using cascaded decoding. The authors proposed a Markov Transformer, which can compute the log-potentials efficiently. The proposed method can be parallelized to get good efficiency during inference. The method gives comparable performance against baselines.

Strengths: Proposed method is general, so it can be applied on many variations of transformer models. Experiment details are clearly described. The proposed method is thoroughly compared with several baselines.

Weaknesses: The clarity, especially the method part needs to be improved. (See Clarity). Since the proposed method does not outperform other models in BLEU, it's necessary to have more discussion. For example, why other methods can not reach same level of parallelism.

Correctness: The method is likely to be correct.

Clarity: The explanation to the method (Sec. 3 and 4) is difficult to understand. I'm not very familiar with non-autoregressive text generation, so I don't fully understand how the generation process works. For example, figure one explains how to get K size-3 spans, but it's unclear how to decode a full sentence using these 3-spans. I would suggest to give a real example on how the algorithm generates a sentence. Notations are not clear. - In line 92, X(x_{i:j}) represents "set of sequences that contain a span x_{i:j}", it's unclear to me if it should appear at the same position or not. - What are C, P and S in TreeMM? Line 100 "wih" -> "with"

Relation to Prior Work: Yes.

Reproducibility: Yes

Additional Feedback: After the authors clarified a few questions in the response, I read the paper again. I feel the algorithm is clear and interesting. But I didn't give a score higher than 6 mainly because the method sections need to be improved.


Review 4

Summary and Contributions: Main Contribution: This paper proposes a cascaded decoding approach with a Markov transformer architecture. The authors also apply treeDP on the decoding process for higher decoding speed, which supports computation in parallel. The cascaded approximation can be computed in parallel in O(MK^2logL).

Strengths: Strength: The proposed idea is novel. Motivated by semi-autoregressive training, the authors propose a Markov transformer architecture and a new cascaded decoding approach. The usage of TreeDP is natural and elegant. Compared with AT-based decoding approaches, the speedup performance is significant. The proposed approach achieves large speedup within little bleu decrease. The related work section is detailed. The paper is in well written.

Weaknesses: Missing some well-performing NAT baselines in experiments. Although the authors list sufficient work in Section 2, the empirical comparison in experiment is limited. Many recent baselines are missed. Compared with NAT-based decoding approaches, the performance improvement is marginal. For example, Levenshtein achieves higher results than the proposed model with similar speedup performance. Also, some NAT approaches have great potential to accelerate decoding, like imitate-NAT and NART-DCRF. They can achieve extremely high speedup performance, like 10X and 18X. Although these approaches also bring large BLEU decays, I still have no idea whether the proposed approach can reach higher BLEU scores under such extremely cases.

Correctness: Yes

Clarity: yes

Relation to Prior Work: yes

Reproducibility: Yes

Additional Feedback: Minor: In line 26, the authors claim that “we can explore any level of autoregressive dependencies to achieve a speed/accuracy tradeoff”. It would be better to add a figure like Figure 3b to show the comparison between the proposed approach and literatures. In line 102-109, the details of treeDP are not clear. It is difficult for me to understand the whole process. To my knowledge, the performance of transformer implemented by fairseq (default version) on De-en should be around 34.7, higher than the reported score 34.4. There is a big margin between Table 1 and line 208. ======== I have read and taken into account the rebuttal.

[Author Response · NeurIPS 2020]

**General Comments:** We thank the reviewers for their thoughtful and useful comments.

Density of Presentation: Several reviewers commented on the density of the presentation. We will add more high-level explanations particularly to make cascaded decoding more intuitive (R2, R3, R4). We agree with the general spirit of simplicity and note that the main methods in this work are actually quite basic. The pseudocode presented for cascade decoding is nearly identical to the implementation used, and the Markov transformer can be implemented with only a few modifications to any transformer architecture.

**Specific Comments:**

**R1** -Long-Term Dependencies: It is a good point that there is a tradeoff between better inference and longer term dependencies among outputs. We note that this method is a compromise in this regard between NAT and fully autoregressive. It is true that we lack long-term dependencies of very long-term models (GPT-3); however it is an open-question of whether all these dependencies are required for conditional generation. For example: one might imagine introducing latent variables to model global topics, or use cascaded decoding to prune the search space for a fully autoregressive model. We think these are promising directions for future work.

-Length Prediction: Like most NAT methods, our work needs to predict length beforehand, which we can do with linear or non-linear regression. However, unlike most of those works, our length constraint only needs to be a *maximum length*. For MT we predicted this value, but for other settings with more variable length it could be just set to a large constant.

**R2** -More powerful CRF modeling: For training speed, we use a locally normalized LM (a special case of a CRF). It is a very interesting suggestion to instead train a globally normalized CRF. This would require running the cascade decoding at training (a difficult but perhaps feasible approach). We think this is interesting future work.

-Relationship to Sun et. al. 2019: We will elaborate on the relationship in our next version. Sun et al introduce a 1st-order CRF on top of a non-autoregressive model. Our work goes beyond a 1st-order CRF and approximates argmax from higher-order CRFs. In addition, we use a single Markov transformer to parameterize all log potentials, instead of using additional side-parameters for pairwise potentials. Lastly, we propose tree decoding to make the parallel complexity sublinear, whereas Sun et al's work is linear.

-Why max marginals and not ngram scores: The problem with using ngram scores is that they do not consider compatibility with other positions. Max-marginal fixes this issue with negligible extra time. On WMT14 En-De val, using ngram scores would get BLEU of 28.42 at 123.48ms, while using max marginals reaches 29.24 at 128.58ms.

**R3** -Performance of approach: NAT research methods inevitably sacrifice accuracy for speedups vs fully autoregressive models with serial search. Compared to these methods, we present a new point on the Pareto frontier of accuracy vs speed: many of our results are very close in accuracy while achieving major speedups. Within this space, we believe these numbers are valuable, particularly given that this is a novel and clean approach: for example, while Levenshtein gets a bit better performance on some datasets, it requires hand-crafted policies for training, and additional tricks at inference. On the other hand, our approach requires much less tuning: at training time we simply used the same set of hyper-parameters as training language models, while tuning $K$ and iters only happens at inference (like beam size in beam search).

-How to generate beyond length 3: We run a parallel version of the forward-backward algorithm, each time pruning low-scoring ngram choices at each position, and we gradually increase n from 1 to $M$. Finally, to generate an entire sentence, we use dynamic programming (Viterbi) to find the sequence with the highest score under the pruned search space. We presented a few examples in appendix A which might be more intuitive.

-Whether $x_{i:j}$ in $\mathcal{X}(x_{i:j})$ should appear at the same position or not: It needs to appear at the same position.

-Tree dynamic programming notations: We will clarify our notations in our next version. $C^i_{j,k_1,k_2}$ is the max possible score of all spans from the $(j * 2^i + 1)$-th position to the $(j + 1) * 2^i$-th position (the length of any span is $2^i$), with the constraint of the left end being word $k_1$ and the right end being word $k_2$. We compute $C^i$ bottom-up, starting from $i = 0$ and merge adjacent spans in $C^i$ to get $C^{i+1}$ (such that span length becomes $2 * 2^i = 2^{i+1}$). The *prefix* score $P^i_{j,k_1,k_2}$ stores the max possible score of all spans from the *1st* position to the $(j * 2^i + 1)$-th position (also with end constraints), which we compute iteratively top-down using $P^{i+1}$ and $C^{i+1}$. Conversely, the *suffix* score $S^i_{j,k_1,k_2}$ is the max score of all spans from the $(j + 1) * 2^i$-th position to the *last* position, also computed top-down. Eventually, we use the prefix scores $P^0$, suffix scores $S^0$, and log potentials $C^0$ to calculate max marginals of any edge.

**R4** -Performance: Please see comment above for R3. We will also include more recent papers in the next version.

-MT Baselines: We followed the exact settings in fairseq translation examples. The 0.3 BLEU score difference might be due to randomness. Our models use the same machine and settings as baselines, so the comparisons aim to be fair. Our IWSLT14 performance baseline is already much better than existing NAT works.

[Meta-Review · NeurIPS 2020]

This paper proposes a semi-autoregressive neural text generation method with cascased transformer where candidate outputs are pruned during decoding through scoring with CRF models of increasing context length. The method supports parallel computing in inference which lead to 7x speed compared to autoregressive method with loss, which is better than most existing non-autoregressive methods. Reviewers all agree that the idea is novel and well develolped. The experiments are extensive and show significant benefit over existing auto-regressive and non-autoregressive methods. It's very nice paper.